# Serum Cholesterol Levels and Risk of Cardiovascular Death: A Systematic Review and a Dose-Response Meta-Analysis of Prospective Cohort Studies

**DOI:** 10.3390/ijerph19148272

**Published:** 2022-07-06

**Authors:** Eujene Jung, So Yeon Kong, Young Sun Ro, Hyun Ho Ryu, Sang Do Shin

**Affiliations:** 1Department of Emergency Medicine, Chonnam National University Hospital, Gwangju 61469, Korea; cnuheujene@gmail.com; 2Laerdal Medical, 4002 Stavanger, Norway; soyeon.kong@gmail.com; 3Department of Emergency Medicine, Seoul National University Hospital, Seoul 03080, Korea; cnuheujene1@gmail.com; 4Department of Emergency Medicine, Chonnam National University, Gwangju 61469, Korea; em.ryu.hyunho@gmail.com

**Keywords:** cholesterol, HDL, LDL, cardiovascular death

## Abstract

Introduction: Numerous studies have demonstrated that abnormal levels of cholesterol are associated with a high attributable risk for the occurrence of cardiovascular disease (CVD). However, there has been no comprehensive study to investigate the relationship between serum cholesterol levels and cardiovascular mortality. Therefore, we conducted a systematic review and dose-response meta-analysis. Methods: A systematic literature search of key databases, including EMBASE and MEDLINE, was conducted and included all the published epidemiological studies that contained estimates of the hazard ratios (HR) of serum cholesterol of CVD mortality. Data extraction, eligibility, and assessment of the risk of bias were assessed by two reviewers independently. All published risk estimates were hazard ratios and analyzed by quantitative meta-analysis using a random-effects model and dose-response relationships of serum cholesterol with CVD mortality. Results: A total of 14 independent reports, including 1,055,309 subjects and 9457 events, were analyzed. The pooled HR (95% CI) was 1.27 (95% CI, 1.19–1.36) for total cholesterol, 1.21 (95% CI, 1.09–1.35) for low-density lipoprotein cholesterol (LDL-C), and 0.60 (95% CI, 0.50–0.72) for high-density lipoprotein cholesterol (HDL-C). We observed a linear association between serum cholesterol (TC, HDL-C) levels and CVD mortality in this meta-analysis. Conclusions: Serum total cholesterol and LDL-C level is associated with increased CVD mortality, but HDL-C level is inversely associated with CVD mortality.

## 1. Introduction

Globally, about 17 million people die from cardiovascular disease (CVD) every year, accounting for 31% of all deaths worldwide [1]. Age-adjusted CVD mortality rates are decreasing in developed countries, but cardiovascular-associated disease remains the leading cause of death due to population aging [2].

A worldwide study demonstrated that among all modifiable risk factors of CV disease, abnormal serum levels of cholesterol were associated with the highest attributable risk for the occurrence of CVD, especially ischemic heart disease [3,4].

High serum total cholesterol (TC) is regarded by many as the main cause of coronary atherosclerosis, and it has been well established that elevated TC is associated with an increased risk of CVD [5]. A lipoprotein is a biochemical assembly whose primary function is to transport fat molecules in water, as in blood plasma or other extracellular fluids; there are High-density lipoprotein cholesterol (HDL-C), Low-density lipoprotein cholesterol (LDL-C), and Very low-density of lipoprotein (VLDL) [6]. LDL-C has been identified as the main risk factor for CVD by many epidemiological and interventional studies because LDL-C plays a major role in the pathogenesis of atherosclerosis [7,8]. Recently, LDL-C has generally replaced TC as the primary lipid measurement for predicting cardiovascular risk. In contrast, a broad body of evidence shows that HDL-C is associated with the risk for vascular complications inversely and is considered an anti-atherosclerotic lipoprotein. There is increasing awareness of the importance of HDL-C in predicting the risk of cardiovascular disease independently of LDL-C levels [9].

Numerous studies have reported the relationship between serum cholesterol and risks of cardiovascular disease and coronary heart disease; however, there has been no comprehensive study to investigate the correlation between the level of serum cholesterol and cardiovascular mortality to our knowledge [5,9]. Additionally, the risk associated with each cholesterol level is uncertain. Understanding the relationship between the level of serum cholesterol and cardiovascular mortality is important for guiding treatment goals for cholesterol levels.

Therefore, we conducted a systematic review and dose-response meta-analysis to determine whether total cholesterol, LDL-C, and HDL-C are risk factors for CVD mortality.

## 2. Materials and Methods

This systematic review and meta-analysis was conducted according to the Preferred Reporting Items for Systematic Reviews and Meta-analysis guidelines [10].

### 2.1. Search Strategy and Selection Criteria

In January 2021, we searched the databases of MEDLINE and EMBASE for cohort studies with human subjects between 2000 and 2020 that had assessed the association between cholesterol and risk of cardiovascular mortality. The computer-based searches focused on the two themes of Medical Subject Headings terms and related exploded versions. The first theme, cholesterol, combined exploded version of the Medical Subject Headings (MeSH) cholesterol, HDL-cholesterol, and LDL-cholesterol. The second theme was heart arrest, cardiovascular death, cardiovascular mortality, cardiac death, or cardiac mortality. The two themes were combined using the Boolean operator “and”.

(Cholesterol or HDL-cholesterol or LDL-cholesterol) and (heart arrest or cardiovascular death or cardiovascular mortality or cardiac death or cardiac mortality).

### 2.2. Study Selection

Studies that met the following criteria were included in this analysis. (1) The study was a prospective cohort study, (2) the exposure was total serum cholesterol, HDL-cholesterol, and LDL-cholesterol levels measured before the cardiovascular event, and (3) the outcome was cardiovascular disease (CVD) mortality, including coronary heart disease (CHD) mortality. We excluded studies if they did not meet these criteria.

### 2.3. Data Extraction and Quality Assessment

One author (E.J.) evaluated the eligibility of the study and data extraction; another author (S.K) reaffirmed the data availability independently. The following data were extracted from each study: year of publication, name of the first author, study design, country of study setting, age, time of follow-up, number of CVD and CHD deaths, number of participants/person-year of follow-up, variable adjusted for in the multivariable analysis, and hazard ratios (HRs). We assessed the methodological quality of the included studies and the risk of bias and advised using elements of the Cochrane collaboration tool to assess the risk of bias [11]. The domains used in the present systematic review belonged to randomization and allocation concealment (selection bias), blinding (performance and detection bias), loss to follow-up, and adherence to the intention to treat principle (attrition bias) [12].

### 2.4. Data Synthesis and Analysis

To analyze the serum level of total cholesterol, LDL-cholesterol, HDL-cholesterol, and risk of CVD death, we used semi-parametric and parametric methods. For the semi-parametric method, we used Comprehensive Meta-Analysis Software 3.0 (Biostat) to analyze the data. In our analysis, we synthesized HRs and 95% CIs to compare the extreme categories of serum cholesterol (highest vs. lowest, as defined within each study). We used the final results of the studies after adjustment for potential confounders from a multivariable model. For the parametric method, a dose-response meta-analysis was performed. The dose-response relationship was estimated by using generalized least-squares trend estimation, according to the methods developed by Greenland and Longnecker [13]. To perform a dose-response meta-analysis, we assigned the midpoint between the upper and lower limits of each category as average consumption. If the upper and lower limits of the category were not provided, both limits were assumed to have the same amplitude as the closet category (the lowest boundary was assumed to be zero, the highest boundary had the same amplitude as the preceding category) [13,14]. The open-source statistical software environment R package (“dosresmeta”) was used for the dose-response analysis.

Forest plots were represented to visually evaluate the HRs and corresponding 95% confidence interval (CIs) throughout the studies. Statistical heterogeneity among the studies was quantified using the Cochrane *Q* statistic (*p* < 0.10 was considered indicative of statistically significant heterogeneity) and the *I*^2^ statistic (ranges from 0% to 100% with lower values representing less heterogeneity) [15]. The *I*^2^ statistic was derived from the *Q* statistic ([*Q* − *df*/*Q*] × 100) and provides a measure of the overall variation attributable to between-study heterogeneity [16]. If no or low heterogeneity (*p* > 0.10) was detected, the HRs were pooled using the fixed-effect model or the DerSimonian and Laird inverse-variance-weighted random-effects models [17]. The funnel plot asymmetry test for measuring publication bias was not used because there were less than 10 studies included in each analysis [18].

Analyses were conducted using Comprehensive Meta-Analysis Software 3.0 (Biostat) and Review Manager Version 5.3 (Cochrane Collaboration). The study is reported according to the PRISMA checklist [19].

## 3. Results

### 3.1. Literature Search

The process of study selection is provided in Figure 1.

The literature search identified 881 studies from PUBMED and 5528 studies from EMBASE. We excluded 690 duplicates, and 5579 articles were deleted after assessing the title or abstract. After reviewing the full articles, another 126 papers were excluded, resulting in a meta-analysis of 14 articles. A manual search of the reference lists of these articles could not find any eligible studies.

### 3.2. Study Characteristics

A total of 14 prospective cohort studies were included in our systematic review and meta-analysis, involving 1,055,309 participants with 9457 CVD deaths or CHD death cases. Of these, 4 studies were conducted in Japanese populations [20,21,22,23], 2 studies in Swedish populations [24,25], 2 studies in Finnish populations [26,27] and 1 study each of Norwegian [28], German [29], Italian [30], Taiwanese, [31] Icelandic [32], and Israeli [33] populations. With the exception of 3 studies [22,27,29], 11 studies had a follow-up duration of more than 10 years [20,21,23,24,25,26,28,30,31,32,33] (Table 1). Cohort studies were assessed by Cochrane Collaboration’s tool for assessing the risk of bias [34] (Figure 2).

### 3.3. Total Cholesterol

#### High versus Low

Eight studies compared the highest level with the lowest (or referent) level categories of total serum cholesterol [23,24,25,27,28,31,32,33] (Figure 3). In 1 study [25] the HR was divided by dividing the age group of 50 years and 70 years, and in another study [32], male and female groups were divided and analyzed, which was considered as a separate study in our study. The study outcomes of 6 studies were CVD mortality [24,25,27,31,32,33], and 2 studies were CHD mortality [23,28]. The overall results were pooled in each study using a fixed-effects model first and a random effect model when the heterogeneity was moderate or higher. The summary HR was 1.27 (95% CI, 1.19–1.36; Figure 3), with moderate heterogeneity (*I*^2^ = 64.1%; *p* < 0.01).

### 3.4. Dose-Response Analysis

We found a significant log-linear dose-response association between serum total cholesterol level and CVD death (*p* < 0.01) and moderate heterogeneity across studies (Q = 8.95, *I*^2^ = 66.5%) (Figure 4).

### 3.5. High-Density Lipoprotein Cholesterol

#### High versus Low

Seven studies compared the lowest level with the highest (or referent) level categories of HDL-C [20,21,22,26,27,29,33]. The outcomes of 6 studies were CVD death [20,21,22,27,29,33] and 1 study was sudden cardiac death [26]. The overall results were pooled in each study using a fixed-effects model first and a random effect model when the heterogeneity was moderate or higher. The summary HR was 0.56 (95% CI, 0.45–0.70; Figure 5), with moderate heterogeneity (*I*^2^ = 40.0%; *p* < 0.01) (Figure 5).

### 3.6. Dose-Response Analysis

We found a significant linear dose-response association between serum HDL-C level and CVD death (*p* < 0.01) and moderate heterogeneity across studies (Q = 7.15, *I*^2^ = 44%) (Figure 6).

### 3.7. Low-Density Lipoprotein Cholesterol

#### High versus Low

Four studies compared the highest level with the lowest (or referent) level categories of LDL-C [26,27,30,33].The outcomes of 3 studies were CVD death [27,30,33], and 1 study was sudden cardiac death [26], which were aggregated together using a fixed-effect model for each study. The summary HR was 1.25 (95% CI, 1.11–1.41; Figure 7), without evidence of heterogeneity (*I*^2^ = 0%; *p* = 0.66) (Figure 7).

## 4. Discussion

Assessments of the relationship between serum cholesterol (TC, LDL-C, and HDL-C) and incidence of cardiovascular disease (CVD) and CVD mortality have been studied for several decades. However, there is an absence of a meta-analysis on CVD mortality and serum levels of cholesterol, which is routinely referred to as a causal factor in producing CVD and is a target of medical treatment of CVD [35].

We conducted a meta-analysis of cohort studies to evaluate the relationship between serum cholesterol (TC, LDL-C, and HDL-c) and CVD mortality. The results of this meta-analysis demonstrate that TC and LDL-C are associated with an increased risk of CVD mortality (HR 1.24, 95% CI 1.16–1.32/1.21, 95% CI 1.09–1.35, respectively), and HDL-C is inversely associated with a risk of CVD mortality. In a dose-response analysis, the relationship between cholesterol (TC, HDL-C) and CVD mortality was linearly associated. Importantly, only population-based studies were included in our study, so these results provide the best available estimates of cholesterol risk in the general population.

Elevated TC is a well-documented and established risk factor for cardiovascular disease. One large meta-analysis based on observational studies found that high levels of serum TC were associated with an increased coronary heart disease mortality rate [36]. The Multiple Risk Factor Intervention Trial (MRFIT) found an association between high serum TC levels and the incidence of ischemic stroke [37]. In an observational study of patients with CHD, TC levels were associated with the risk of ischemic stroke (RR, 1.43; 95% CI, 1.20–1.70), whereas another study in patients without CHD did not show an association with serum lipids [38]. However, several studies revealed that elevated TC is not associated with CVD, and TC is not the most accurate predictor of future CVD [39,40]. Thus, in recent studies about serum cholesterol, the interest in the role of LDL-C is increasing. There are many studies that show that LDL-C, one of the cholesterols constituting TC, causes atherosclerosis-related disease. However, according to a comprehensive review of the association of LDL-C with atherosclerosis, some studies have shown a lack of an association between LDL-C and severity of atherosclerosis, and in a study of 304 women, there was no association between LDL-C and coronary calcification [41,42].

Despite several conflicting studies, total cholesterol and LDL-C are key components of cardiovascular risk prediction models that are widely used in clinical practice to estimate an individual’s risk of CVD and to guide clinical decision-making regarding the initiation of statin therapy and other lipid-level regulating drugs [43]. In studies of the relationship between cholesterol and CVD occurrence, treatment with TC or LDL-C lowering drugs reduces CVD mortality [44,45]. In our meta-analysis, TC and LDL-C are risk factors for CVD death, which supports previous studies suggesting that maintaining adequate levels of cholesterol is important in lowering CVD death risk [46].

On the contrary, we observed that high levels of HDL-C were significantly associated with decreased risk of CVD mortality. The first compelling study of the inverse association between HDL-C and CHD was from the Framingham Heart Study [47]. This observational study formed the basis for the widely acknowledged concept of HDL-C as the good cholesterol and led to the idea that HDL-C might have properties that protect against atherosclerosis, and intervention to increase HDL-C would reduce CHD risk (the HDL-C hypothesis). This hypothesis was supported by a series of animal studies and observation studies after the Framingham reports [48,49,50,51]. However, some studies suggested that very high HDL-C levels are no longer effectively protective, which is removing cholesterol from the periphery. Also, the cholesterol content of HDL does not represent many important HDL functions that are related to CVD risk, such as anti-inflammatory, antiapoptotic, antioxidant, and vasorelaxant properties.

Previous studies have shown that levels and normal ranges of cholesterol vary by age, gender, and race, accordingly, and the level of cholesterol as a risk factor for disease may be different. Studies included in our meta-analysis targeted patients of various ages. In one study, they studied patients aged 50 and 70 separately [25]; in other studies, only the age range of the patients was presented [26,27,31]. In the remaining studies [20,21,22,23,24,25,28,29,30,32,33], the mean age of the subjects was presented, and the mean age ranged from 42.1 [33] to 73.8 [30] years old. The gender of the study subjects showed various ratios in each study. In one study, males and females were analyzed separately [32]. Five studies were conducted on Asians [20,21,22,23,31], and the rest were on Europeans. In addition, the observation period of the cohort varied from 7.8 [27] to 33 [28] years. Two studies divided into two groups according to age and gender did not coincide with each other and were regarded as two different studies [25,32].

As we know, this is the first meta-analysis that explains the relationship between serum cholesterol and CVD mortality. The analysis shows that TC and LDL-C are associated with increased CVD mortality, whereas HDL-C is inversely associated. Our study may be useful in determining the therapeutic targets of serum cholesterol. Furthermore, there are several studies showing different ideal serum levels of cholesterol by age and gender, so a subgroup meta-analysis of these studies would provide useful information. The main results of the analysis were CVD mortality, and the study used it when CHD mortality was presented as a study outcome [23,28]. One study, in which the study outcome was sudden death, was included in the final meta-analysis [26].

In our study, TC showed a linear relationship with CVD mortality. This will serve as a theoretical basis for suggesting target levels of TC to reduce CVD mortality, and further studies will be needed to obtain accurate target levels of cholesterol according to patient characteristics in the future.

### Limitations of the Study

We are aware of several potential limitations. First, measured LDL-C also comprises the fraction that is transported with lipoprotein(a); we did not take this fact into consideration. Second, studies with meaningful results may be more likely to be published, and are preferentially published in English journals [52]. Third, although adjusted estimates were reported in general, the observational design had its inherent weak point that any association may be because of the presence of inapposite measured or unmeasured residual possible confounding variables. Fourth, as noted above, the risk of bias in index studies meant that evidence quality in relation to study outcomes was categorized as very low. Fifth, our decision to meta-analysis data may be questioned, given the marked heterogeneity between each study. However, we noted that overall statistical heterogeneity for each meta-analysis, as measured by the *I*^2^ statistic, was moderate, and we chose a random-effect model to account for differences in effect size between each study [53]. Sixth, all studies included in our meta-analysis provided no information about lipid-level regulating medication that affected overall study results. Seventh, in some studies, different HR values according to age and sex were reported, but our study could not report it because the number of studies was too small, and the heterogeneity between studies was high. Lastly, in general, the serum LDL-C level is not measured directly but instead is estimated from its cholesterol concentration. The LDL-C-related studies included in our meta-analysis did not address the method of direct or indirect calculating of LDL-C levels, which may have affected the outcomes.

## 5. Conclusions

In our meta-analysis, high serum levels of TC and LDL-C increased the CVD mortality above the reference level, and low HDL-C was associated with an increased CVD mortality. We believe that this study will be helpful in setting appropriate cholesterol management targets, and personal characteristics, such as age and gender, are thought to influence the ideal serum cholesterol level, so further research is needed.

## Figures and Tables

**Figure 1 ijerph-19-08272-f001:**
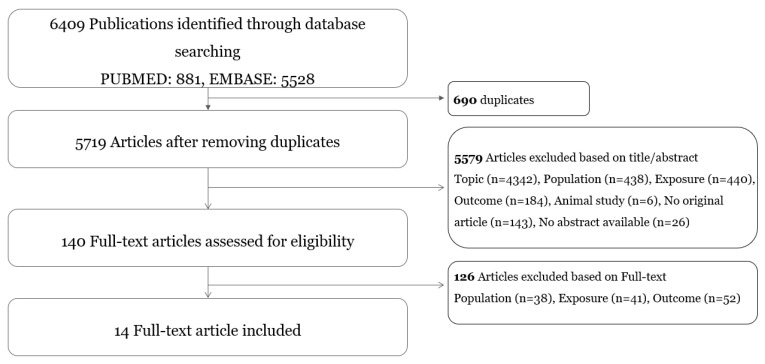
Flowchart of the study selection.

**Figure 2 ijerph-19-08272-f002:**
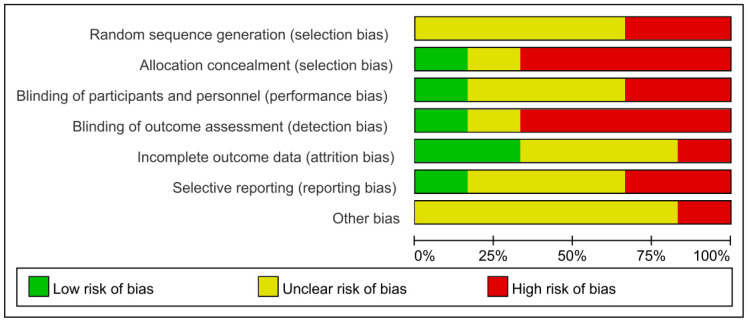
Cochrane Collaboration’s tool for assessing the risk of bias.

**Figure 3 ijerph-19-08272-f003:**
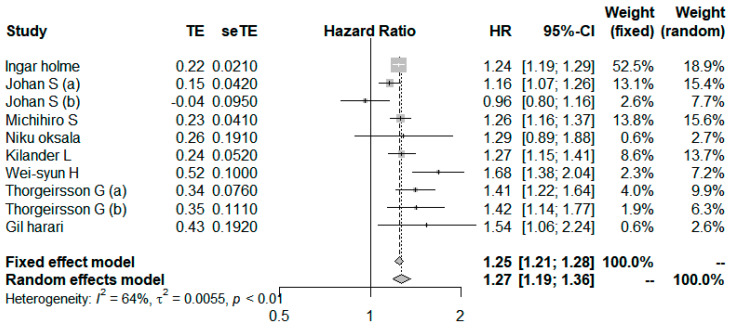
Forest plot for pooled HRs and 95% CIs of CVD mortality for highest versus lowest categories of Serum Total cholesterol (TC) level [23,24,25,27,28,31,32,33]. The overall effect was obtained from a random-effect model.

**Figure 4 ijerph-19-08272-f004:**
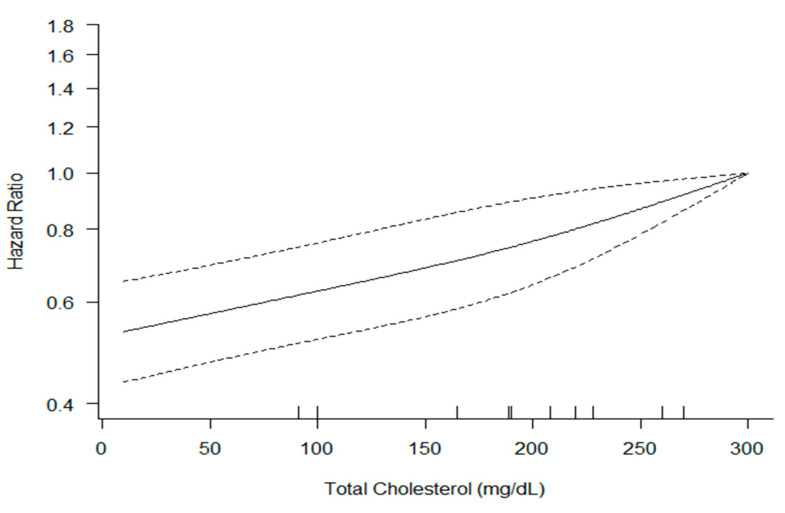
Dose-response meta-analysis between serum total cholesterol level and the hazard ratio of CVD mortality. The solid line represents point estimates of the association of serum total cholesterol and CVD mortality with the use of restricted cubic splines models, and the dashed lines indicate 95% CIs.

**Figure 5 ijerph-19-08272-f005:**
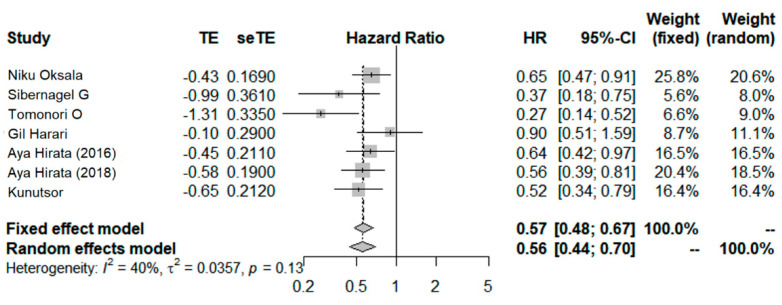
Forest plot for pooled HRs and 95% CIs of CVD mortality for lowest versus highest categories of serum High-density lipoprotein cholesterol (HLD-C) level [20,21,22,26,27,29,33]. The overall effect was obtained from a random-effect model.

**Figure 6 ijerph-19-08272-f006:**
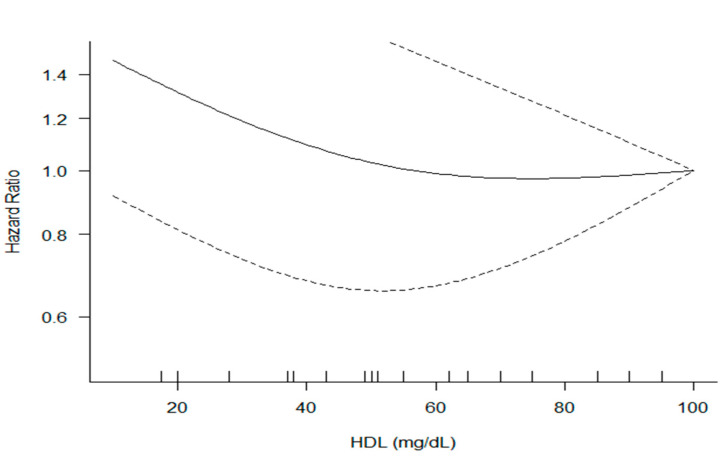
Dose-response meta-analysis between serum High-density lipoprotein (HDL) cholesterol level and the hazard ratio of CVD mortality. The solid line represents point estimates of the association of serum HDL cholesterol and CVD mortality with the use of restricted cubic splines models, and the dashed lines indicate 95% CIs.

**Figure 7 ijerph-19-08272-f007:**
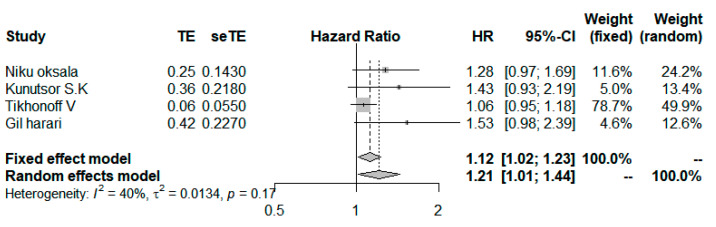
Forest plot for pooled HRs and 95% CIs of CVD mortality for highest versus lowest categories of serum Low-density lipoprotein cholesterol (LDL-C) level [26,27,30,33]. The overall effect was obtained from a random-effect model.

**Table 1 ijerph-19-08272-t001:** Characteristics of included cohort studies of serum cholesterol level and CVD mortality.

Study (Author, Year)	Country	Duration of Follow-Up (Years)	Age (Years)	Total N (Cases)	Setting	Clinical Endpoints	Effect Size (HR, 95% CI)
Ingar Holme (2011) [28]	Norway	33	45	14,846 (1655)	Oslo study	CHD mortality	<TC>: 1.24 (1.19–1.29)
Johan S (2006) [25]	Sweden	32.7	50 70	2841 (1078) 168 (302)	Uppsala County	CVD mortality	<TC> age 50: 1.16 (1.07–1.26) age 70: 0.96 (0.80–1.16)
Michihiro. S (2015) [23]	Japan	15	56.9	73,916 (770)	EPOCH-JAPAN group	CHD mortality	<TC>: 1.26 (1.16–1.34)
Nicu Oksala (2012) [27]	Finland	7.8	30–65	5956 (55)	Health 2000 survey	SCD	<TC>: 1.29 (0.89–1.88) <LDL>: 1.28 (0.97–1.70) <HDL>: 0.65 (0.47–0.91)
Kilander. L (2001) [24]	Sweden	25.7	50	2301 (301)		CVD mortality	<TC>: 1.27 (1.15–1.41)
Silbernagel. G (2013) [29]	Germany	8.9	62.6	3141 (590)	LURIC study	CVD mortality	<HDL>: 0.37 (0.18–0.74)
Tomonori. O (2005) [22]	Japan	9.6	52.8	7175 (174)	The Nippon General DATA 90 research group	CVD mortality	<HDL>: 0.27 (0.14–0.52)
Kunutsor. S. K (2017) [26]	Finland	23	42–61	2616 (228)	KIHD risk factor study	Sudden death	<LDL>: 1.43 (0.93–2.19) <HDL>: 0.52 (0.34–0.78)
Tikhonoff. V (2005) [30]	Italy	12	73.8	3120 (327)		CVD mortality	<LDL>: 1.06 (0.94–1.18)
Wei-Syun. H (2017) [31]	Taiwan	8.8	Over 20	381,963 (1894)	The MJ health group	CVD mortality	<TC>: 1.68 (1.38–2.04)
Thorgeirsson. G (2005) [32]	Iceland	25	M: 66.8 W: 65.8	8806 (137) 9435 (44)	The Reykjavik study	CVD mortality	<TC> M: 1.41 (1.21–1.63) W: 1.42 (1.14–1.76)
Gil Harari (2017) [33]	Israel	22	42.1	4832 (172)	CORDIS	CVD mortality	<TC>: 3.58 (2.49–5.14) <LDL>: 1.82 (1.15–2.89) <HDL>: 0.90 (0.51–1.59)
Aya Hirata (2016) [20]	Japan	18	52.5	7019 (450)	NIPPON DATA study	CVD mortality	<HDL>: 0.64 (0.42–0.96)
Aya Hirata (2018) [21]	Japan	12.1	57.1	525,661 (1280)	EPOCH-JAPAN group	CVD mortality	<HDL>: 0.56 (0.39–0.82)

CHD = coronary heart disease; CVD = cardiovascular disease; HR = hazard ratio; TC = Total cholesterol; HDL = high density lipoprotein; LDL = low density lipoprotein.

## Data Availability

Not applicable.

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
