# Peer review of "Serum Cholesterol Levels and Risk of Cardiovascular Death: A Systematic Review and a Dose-Response Meta-Analysis of Prospective Cohort Studies"

_ijerph, 2022, doi:10.3390/ijerph19148272_

Round 1

Reviewer 1 Report

It is an interesting study, but grammar must be improved. References in the section of references are not matching with tables and figures. I could not find the articles. 

1. Grammar must be improved

1a. Line 42 to 45, this paragraph is not clear. It must be rewritten

2a. Line 47 to 49 references have to be homogenized 

3a. Period must be places at the end of the reference parenthesis.

2. Introduction

2a. Line 56-57 which numerous studies. Examples must be included.

2b. Objective is not clear. Why do you mean by relationship? Do you mean cholesterol as risk factor for mortality?

3. Methodology

3a. Studies were searched between what years. Please specify.

3b Line 73 search was conducted in January 2021, but same point as 3a. It includes articles from what years?

3c. This search included animal trials as well? 

3d. It was adjusted for age and sex as well?

4. Results

4a. In general, results must be rewritten. it is not clear all results section

4b. Section 3.3.3 What means "outcomes of 6 studies were CVD dears and 2 studies were CHD death." It is not clear

4c. Maybe important to include Race and sex in Table 1.

4d. References in Tables and Figures do not match the list of references. For example Ingar Holme (2011). Is Holme and Tonstad, 2013? or which one is it?

Johan S. (2006) Which one is this paper? It is difficult to compare references with this present manuscript. 

5. Discussion

5a. Why do you mean by absence  of a literature review.There are more similar reviews with this association (Ranvskov et al., 2016 BMJ and Kwon et al 2019)

5b. I would like to extend the discussion regarding the linear association responses in the discussion  

5c. I would like to see recommendation for future clinical trials characteristics to improve meta-analysis 

Author Response

It is an interesting study, but grammar must be improved. References in the section of references are not matching with tables and figures. I could not find the articles. 

  1. Grammar must be improved

1a. Line 42 to 45, this paragraph is not clear. It must be rewritten

Answer: Thank you for valuable comments. We revised sentence for your advice

Line 42-45: A lipoprotein is biochemical assembly whose primary function is to transport fat molecules in water, as in blood plasma or other extracellular fluids, and there are High-density lipoprotein cholesterol (HDL-C), Low-density lipoprotein cholesterol (LDL-C), and very low-density of lipoprotein (VLDL).(Gofman et al., 1954)

2a. Line 47 to 49 references have to be homogenized 

Answer: Thank you for valuable comments. We revised references for your advice

Line 48: (Steinberg, 2009; Wilson et al., 1998)

3a. Period must be places at the end of the reference parenthesis.

Answer: Thank you for valuable comments. We corrected the position of the Period in the entire manuscript according to your advice.

  1. Introduction

2a. Line 56-57 which numerous studies. Examples must be included.

Answer: Thank you for valuable comments. We added reference according to your advice.

Line 56: (Stamler et al., 2000; Gordon et al., 1977)

2b. Objective is not clear. Why do you mean by relationship? Do you mean cholesterol as risk factor for mortality?

Answer: Thank you for valuable comments. We clarified the sentence for your advice.

Line 60-61: So, we conducted a systematic review and dose-response meta-analysis to whether total cholesterol, LDL-C and HDL-C are risk factors for CVD mortality.

  1. Methodology

3a. Studies were searched between what years. Please specify.

Answer: Thank you for valuable comments. We added analyzed period.

Line 67-68: between 2000 and 2020

3b Line 73 search was conducted in January 2021, but same point as 3a. It includes articles from what years?

Answer: Thank you for valuable comments. We added analyzed period.

Line 67-68: between 2000 and 2020

3c. This search included animal trials as well? 

Answer: Thank you for valuable comments. We analyzed only the cohort study with human subjects, and the related comments were additionally described

Line 67-68: We first searched the databases of MEDLINE and EMBASE for cohort studies with human subjects between 2000 and 2020

3d. It was adjusted for age and sex as well?

 Answer: In all studies, in common, age and sex were adjusted.

  1. Results

4a. In general, results must be rewritten. it is not clear all results section

Answer: Thank you for valuable comments. We rewrote Results section according to your advice.

4b. Section 3.3.3 What means "outcomes of 6 studies were CVD dears and 2 studies were CHD death." It is not clear

Answer: Thank you for valuable comments. We revised this sentence according to your advice.

Line 164-167: The study outcomes of 6 studies were CVD mortality(Harari et al., 2017; Hu et al., 2017; Kilander et al., 2001; Oksala et al., 2013; Sundstrom et al., 2006; Thorgeirsson et al., 2005) and 2 studies were CHD mortality(Holme & Tonstad, 2013; Satoh et al., 2015).

4c. Maybe important to include Race and sex in Table 1.

Answer: Thank you for your valuable comments. In most studies, country of study was presented instead of race, and studies that differentiated sex are presented in the Table 1.

4d. References in Tables and Figures do not match the list of references. For example Ingar Holme (2011). Is Holme and Tonstad, 2013? or which one is it?

Johan S. (2006) Which one is this paper? It is difficult to compare references with this present manuscript. 

Answer: Thank you for your comments. We revised references according to your advice.

  1. Discussion

5a. Why do you mean by absence  of a literature review.There are more similar reviews with this association (Ranvskov et al., 2016 BMJ and Kwon et al 2019)

Answer: Thank you for your comments. There have been some systematic reviews showing the association between cholesterol and CVD mortality, but there was no meta-analysis like ours as we know. This has been clearly commented.

Line 225-226: there is an absence of a meta-analysis on CVD mortality and serum levels of cholesterol

5b. I would like to extend the discussion regarding the linear association responses in the discussion  

5c. I would like to see recommendation for future clinical trials characteristics to improve meta-analysis 

Answer: Thank you for your valuable comments. We added some sentences about linear association and future clinical trials.

Line 300-303: In our study, TC and HDL-C showed a linear relationship with CVD mortality. This will serve as a theoretical basis for suggesting target levels of TC and HDL-C to reduce CVD Mortality, and further studies will be needed to obtain accurate target levels of Cholesterol according to patient characteristics in the future.

Reviewer 2 Report

The authors analyzed the relationship between serum cholesterol level and cardiovascular mortality. The conclusions drawn for total cholesterol and LDL-cholesterol are correct. The interpretation of the effect of high HDL-C levels should be interpreted somewhat otherwise – studies with CETP inhibitors where HDL-C was increased did not show any positive effect on CVD.

It should be taken into consideration that a part of the measured LDL-C is transported with Lipoprotein(a) – this fact should at least be mentioned.

Question: Have studies been published analyzing the relationship between Apolipoprotein B concentrations and CVD mortality?

Minor comments

Line 43: triglyceride – usually “triglycerides” is written

Line 142: included our systematic review – “into” is lacking

Figore 2: it is not clear whether this assessment tool means the intention or the measured result

Table 1: it is astonishing that the HR for HDL is 1.32 in the Israel study (moreover the range is rather wide – the upper 95% is 2.27 – here something may be wrong); in the other studies this limit is below 1; in Figure 5 other data are given for this study

Figure 3: Niku oksala – “Oksala”; Wei-syun H – “Syun”; Study Gil Harari (big H at the beginning of Harari)

Figure 4: the reader does not understand why Hazard Ratio is below 1 even in higher TC levels – the HRs given in Figure 3 are all above 1 (with the exception of Johan S (b) and Nku Oksala)

Figure 5: in the caption the authors compare lowest versus highest categories, in the text the highest versus the lowest are compared – please check

Line 218: and 1 studies – not correct English

Figure 7: Gil harari – should read Harari

Line 248: several studies revealed that elevated TC is no associated with CVD – write “is not associated”

Line 257: total cholesterol and LDL-C is a key component – write “are key components”

Line 263: TC and LDL-C is risk factor of CVD death – write “are risk factors for CVD death”

Line 266: On the contrary, we observed that high levels of HDL-C were significantly associ-ated with decreased risk of CVD mortality. – in fact, very high HDL-C levels are no longer protective against CVD – it was described that HDL-C levels do not reflect the function of HDL – that is removing cholesterol from the periphery; moreover extremely high HDL concentrations may point to an altered HDL composition

Line 274: As described above, studies on the relationship between serum cholesterol level and atherosclerosis-related disease such as CHD and ischemic stroke. – this sentence should be deleted

Line 278: according to age, gender and rate may be – these words are given twice in this sentence – please correct (also race instead of rate)

Line 281: studied patients age 50 and 70 separately – write “aged” instead of “age”

Line 324: high HDL-C decreased the CVD mortality. – the reviewer thinks that this conclusion is not correct – it should read: low HDL-C levels are associated with an increased CVC mortality – see comments to Line 266

Author Response

The authors analyzed the relationship between serum cholesterol level and cardiovascular mortality. The conclusions drawn for total cholesterol and LDL-cholesterol are correct. The interpretation of the effect of high HDL-C levels should be interpreted somewhat otherwise – studies with CETP inhibitors where HDL-C was increased did not show any positive effect on CVD.

It should be taken into consideration that a part of the measured LDL-C is transported with Lipoprotein(a) – this fact should at least be mentioned.

Answer: Thank you for your valuable comments. We added some sentence in limitation section according your advice.

Line 307-308: First, measured LDC-C is transported with Lipoprotein, however, it did not considered.

Question: Have studies been published analyzing the relationship between Apolipoprotein B concentrations and CVD mortality?

Answer: Thank you for valuable comments. No studies have investigated Apolipoprotein B and CVD mortality, although there are some studies that have reported the association Apolipoprotein B and CVD incidence.

Minor comments

Line 43: triglyceride – usually “triglycerides” is written

Answer: Thank you for valuable comments. We corrected sentence according to your advice.

Line 142: included our systematic review – “into” is lacking

Answer: Thank you for valuable comments. We corrected sentence according to your advice.

Figore 2: it is not clear whether this assessment tool means the intention or the measured result

Answer: Thank you for valuable comments. Cochrane Collaboration’s tool for assessing risk of bias is a summary of the values measured by the authors based on the criteria presented in previous studies.

Table 1: it is astonishing that the HR for HDL is 1.32 in the Israel study (moreover the range is rather wide – the upper 95% is 2.27 – here something may be wrong); in the other studies this limit is below 1; in Figure 5 other data are given for this study

Answer: Thank you for valuable comments. We revised Table 1 according to your advice.

Table 1: <HDL> 0.90 (0.51-1.59)

Figure 3: Niku oksala – “Oksala”; Wei-syun H – “Syun”; Study Gil Harari (big H at the beginning of Harari)

Answer: Thank you for valuale comments. We revised Figure 3 according to your advice.

Figure 4: the reader does not understand why Hazard Ratio is below 1 even in higher TC levels – the HRs given in Figure 3 are all above 1 (with the exception of Johan S (b) and Nku Oksala)

Answer: Thank you for valuble comments. We suggest log-linear dose-response association between serum total cholesterol and CVD mortality.

Figure 5: in the caption the authors compare lowest versus highest categories, in the text the highest versus the lowest are compared – please check

Answer: Thank you for valuable comments. We revised text according to your advice.

Line 187: Seven studies compared the lowest level with highest (or referent) level categories of HDL-C

Line 218: and 1 studies – not correct English

Answer: Thank you for valuable comment. We revised sentence according to your advice.

Line 214: and 1 study was

Figure 7: Gil harari – should read Harari

Answer: Thank you for valuale comments. We revised Figure 7 according to your advice.

Line 248: several studies revealed that elevated TC is no associated with CVD – write “is not associated”

Answer: Thank you for valuable comments. We revised words according to your advice.

Line 257: total cholesterol and LDL-C is a key component – write “are key components”

Answer: Thank you for valuable comments. We revised words according to your advice.

Line 263: TC and LDL-C is risk factor of CVD death – write “are risk factors for CVD death”

Answer: Thank you for valuable comments. We revised words according to your advice.

Line 266: On the contrary, we observed that high levels of HDL-C were significantly associ-ated with decreased risk of CVD mortality. – in fact, very high HDL-C levels are no longer protective against CVD – it was described that HDL-C levels do not reflect the function of HDL – that is removing cholesterol from the periphery; moreover extremely high HDL concentrations may point to an altered HDL composition

Answer: Thank you for valuable comments. We added some sentences according to your advice.

Line 269-271: However, some studies suggested that very high HDL-C levels are no longer protective effect, which is removing cholesterol from the periphery.

Line 274: As described above, studies on the relationship between serum cholesterol level and atherosclerosis-related disease such as CHD and ischemic stroke. – this sentence should be deleted

Answer: Thank you for valuable comments. We deleted some sentence according to your advice.

Line 278: according to age, gender and rate may be – these words are given twice in this sentence – please correct (also race instead of rate)

Answer: Thank you for valuable comments. We revised sentence according to your advice.

Line 272-274: Previous studies have shown that levels and normal ranges of cholesterol vary by age, gender and race, accordingly, and the limit level of cholesterol as a risk factor of disease may be different

Line 281: studied patients age 50 and 70 separately – write “aged” instead of “age”

Answer: Thank you for valuable comments. We revised words according to your advice.

Line 324: high HDL-C decreased the CVD mortality. – the reviewer thinks that this conclusion is not correct – it should read: low HDL-C levels are associated with an increased CVC mortality – see comments to Line 266

Answer: Thank you for valuable comments. We revised sentence according to your advice.

Line 325-326: Previous studies have shown that levels and normal ranges of cholesterol vary by age, gender and race, accordingly, and the limit level of cholesterol as a risk factor of disease may be different

Reviewer 3 Report

INTRODUCTION. The presentation of the different cholesterol fractions and pof their relationship with CV risk is somehat confused, and referencesa are very old. Please, see Ference BA, et al. Eur Heart J. 2017 Aug 21;38(32):2459-2472 and Casula M, et al. 2021 Jul 23;10(8):1869. 

Authors present their paper as the first regarding the correlation between level of serum cholesterol 58 and cardiovascular mortality. However, this issue has already been addressed (see Khan SU, et al. Am J Prev Cardiol. 2020;1:100013; Navarese EP, et al. JAMA. 2018;319(15):1566–1579; Abdullah SM, et al. Circulation. 2018 Nov 20;138(21):2315-2325; Madsen CM, et al. Eur Heart J201738:2478–2486)

The type of included studies (observationa studies) should be clearly stated in the title and in the aim.

METHODS. Please provide as supplementary material an example of the search strategy in one of the database.

RESULTS. In general, I am not sure that the research was exhaustive. The topic addressed by the authors is extremely discussed in the literature, while few studies were found. Furthermore, there are studies that have not been included, see e.g. Johannesen CDL, et al. BMJ. 2020 Dec 8;371:m4266; or Abdullah SM, et al. Circulation. 2018 Nov 20;138(21):2315-2325.

Authors siad that quality was assessed through Newcastle-Ottawa scale, while Figure 2 reported the Cochrane Collaboration’s tool for assessing risk of bias, as stated in Methods.

The sentence "The overall results were pooled in each study using a fixed effects model first, which were aggregated into the analysis, using a random effect model." is not clear. Please, clarify.

DISCUSSION. Here, as well as in the Introduction, the association between cholesterol and cardiovascular risk is higlighted, but without a specific focus on CV mortality. As so, the aim of the present meta-analysis is not well supported.

The result of this meta-analysis is the quantitave association between cholesterol and CV mortality. However, numbers are not discussed, and the discussion is only at qualitative level.

Discussion of subgroups (e.g. by gender and age) is also not entirely appropriate. Separate analyses within individual studies are cited, but without reporting their results.

Author Response

INTRODUCTION. The presentation of the different cholesterol fractions and pof their relationship with CV risk is somehat confused, and referencesa are very old. Please, see Ference BA, et al. Eur Heart J. 2017 Aug 21;38(32):2459-2472 and Casula M, et al. 2021 Jul 23;10(8):1869. 

Answer: Thank you for valuable comments. We added new reference according to your advice.

References 2: Casula M, Colpani O, Xie S, Alberico L, & Baragetti A (2021). HDL in Atherosclerotic Cardiovascular Disease: In Search of a Role. Cells 201, 10(8)

Authors present their paper as the first regarding the correlation between level of serum cholesterol 58 and cardiovascular mortality. However, this issue has already been addressed (see Khan SU, et al. Am J Prev Cardiol. 2020;1:100013; Navarese EP, et al. JAMA. 2018;319(15):1566–1579; Abdullah SM, et al. Circulation. 2018 Nov 20;138(21):2315-2325; Madsen CM, et al. Eur Heart J. 2017; 38:2478–2486)

Answer: Thank you for valuable comments. As you pointed out, there are a number of studies reporting Cholesterol (TC, HDL, LDL) and CVD mortality, but, meta-analysis is rare, and non of the studies reported a dose-response effect, as we know.

The type of included studies (observationa studies) should be clearly stated in the title and in the aim.

Answer: Thank you for valuable comments. Our study included only prospective cohort study. We mentioned this in the title and method section.

METHODS. Please provide as supplementary material an example of the search strategy in one of the database.

Answer: Thank you for valuable comments. We provided search strategy in method section according to your advice.

Line 78-79: (Cholesterol or HDL-cholesterol or LDL-cholesterol) and (heart arrest or cardiovascular

death or cardiovascular mortaltiy or cardiac death or cardiac mortality)

RESULTS. In general, I am not sure that the research was exhaustive. The topic addressed by the authors is extremely discussed in the literature, while few studies were found. Furthermore, there are studies that have not been included, see e.g. Johannesen CDL, et al. BMJ. 2020 Dec 8;371:m4266; or Abdullah SM, et al. Circulation. 2018 Nov 20;138(21):2315-2325.

Answer: Thank you for valuable comments. In our study, the study outcome was CVD mortality or CHD mortality, so we could not add the study you recommended after discussing with the all authors.

Authors siad that quality was assessed through Newcastle-Ottawa scale, while Figure 2 reported the Cochrane Collaboration’s tool for assessing risk of bias, as stated in Methods.

Answer: Thank you for valuable comments. We revised sentences according to your advice.

Line 155: Cohort studies were assessed by Cochrane Collaboration’s tool for assessing risk of bias

The sentence "The overall results were pooled in each study using a fixed effects model first, which were aggregated into the analysis, using a random effect model." is not clear. Please, clarify.

Answer: Thank you for valuable comments. We revised sentences according to your advice

Line 173-175: The overall results were pooled in each study using a fixed effects model first, and using a random effect model when the heterogeneity was moderate or higher

DISCUSSION. Here, as well as in the Introduction, the association between cholesterol and cardiovascular risk is higlighted, but without a specific focus on CV mortality. As so, the aim of the present meta-analysis is not well supported.

Answer: Thank you for valuable comments. We revised sentences according to your advice.

Line 235-239: The results of this meta-analysis demonstrate that TC and LDL-C is associated with an increased risk of CVD mortality (HR 1.24, 95% CI 1.16-1.32 / 1.21, 95% CI 1.09-1.35, respectively) and HDL-C is inversely associated with a risk of CVD mortality. In dose-response analysis of the relationship between cholesterol (TC, HDL-C) and CVD mortality was linearly associated.

The result of this meta-analysis is the quantitave association between cholesterol and CV mortality. However, numbers are not discussed, and the discussion is only at qualitative level.

Answer: Thank you for valuable comments. In prospective cohort study included in our study, because of the high heterogeneity between studies, it was thought that there was a limit to presenting a numerical association, so quantative association was mainly presented, and a dose-response analysis was used for insufficient results.

Discussion of subgroups (e.g. by gender and age) is also not entirely appropriate. Separate analyses within individual studies are cited, but without reporting their results.

Answer: Thank you for valuable comments. Most of enrolled studies did not provide HR values according to age and sex, so data requests were made to the authors several times, but, the analysis could not be performed. We described in limitation section.

Seventh, in some studies, different HR values according to age and sex were reported, but our study could not report it because the number of studies was too small and the heterogeneity between studies was high.

Round 2

Reviewer 2 Report

The authors satisfactorily answered to my points.

But there remain still some problems:

Figure 5: several names still begin with lowercase letters

In Figures 3, 5, 7 the caption should read: Forest plot for pooled HRs and 95% CIS of CVD mortality for highest versus lowest categories  - instead of (as written by the authros): Forest plot for pooled HRs and 95% CIs of CVD mortality for lowest versus highest categories. Explanation: higher levels of LDL-C or TC are associated with higher HRs, for HDL-C they are associated with lower HRs

Line 275: However, some studies suggested that very high HDL-C levels are no longer protective effect, which is removing cholesterol from the periphery. Also, the cholesterol content of HDL does not represent many infortant HDL functions that are related to CVD risk, such as anti-inflammatory, antiapoptotis, antioxidant, and vasorelaxant properties. – there are several mistakes in these sentences: 1. HDL-C levels are no longer effectively protective, 2. Infortant should read important, 3. anitapoptotis should read “antiapoptotic”

Line 307: In our study, TC and HDL-C showed a linear relationship with CVD mortality. This will Serve as a theoretical basis for suggesting target levels of TC and HDL-C to reduce CVD Mortality, - the reviewer does not agree that there will be a suggested target level for HDL- to reduce CVD mortality – no effective measures to act on HDL-C concentrations are known

Line 314: First, measured LDC-C is transported with Lipoprotein, however, it did not considered. – This sentence is a mess – it should read: First, measured LDL-C comprises also the fraction that is transported with Lipoprotein(a), we did not take this fact into consideration.

References: they should be carefully checked by the authors

Reference 18: Brackets are missing for 2011

Reference 23: no reference is given

In some references  no doi data are given

Author Response

Figure 5: several names still begin with lowercase letters

Answer: Thank you for valuable comment. We revised figure as your advice.

In Figures 3, 5, 7 the caption should read: Forest plot for pooled HRs and 95% CIS of CVD mortality for highest versus lowest categories  - instead of (as written by the authros): Forest plot for pooled HRs and 95% CIs of CVD mortality for lowest versus highest categories. Explanation: higher levels of LDL-C or TC are associated with higher HRs, for HDL-C they are associated with lower HRs

Answer: Thank you for valuable comment. We revised caption of figure as your advice.

Line 275: However, some studies suggested that very high HDL-C levels are no longer protective effect, which is removing cholesterol from the periphery. Also, the cholesterol content of HDL does not represent many infortant HDL functions that are related to CVD risk, such as anti-inflammatory, antiapoptotis, antioxidant, and vasorelaxant properties. – there are several mistakes in these sentences: 1. HDL-C levels are no longer effectively protective, 2. Infortant should read important, 3. anitapoptotis should read “antiapoptotic”

Answer: Thank you for valuable comment. We revised sentences as your advice.

Line 307: In our study, TC and HDL-C showed a linear relationship with CVD mortality. This will Serve as a theoretical basis for suggesting target levels of TC and HDL-C to reduce CVD Mortality, - the reviewer does not agree that there will be a suggested target level for HDL- to reduce CVD mortality – no effective measures to act on HDL-C concentrations are known

Answer: Thank you for valuable comment. In our analysis, a linear relationship between HDL and CVd mortality was observed, but, as you pointed out, the theoretical evidence was lacked, so we did not mention it in the manuscript.

Line 314: First, measured LDC-C is transported with Lipoprotein, however, it did not considered. – This sentence is a mess – it should read: First, measured LDL-C comprises also the fraction that is transported with Lipoprotein(a), we did not take this fact into consideration.

Answer: Thank you for valuable comment. We revised sentences as your advice.

References: they should be carefully checked by the authors

Reference 18: Brackets are missing for 2011

Reference 23: no reference is given

In some references  no doi data are given

Answer: Thank you for valuable comment. We revised references as your advice